# Optimising Regimen of Co-Amoxiclav (ORCA)—The Safety and Efficacy of Intravenous Co-Amoxiclav at Higher Dosing Frequency in Patients with Diabetic Foot Infection

**DOI:** 10.3390/antibiotics14080758

**Published:** 2025-07-28

**Authors:** Jun Jie Tan, Peijun Yvonne Zhou, Jia Le Lim, Fang Liu, Lay Hoon Andrea Kwa

**Affiliations:** 1Singapore General Hospital, Singapore 169608, Singapore; yvonne.zhou.p.j@sgh.com.sg (P.Y.Z.); lim.jia.le@sgh.com.sg (J.L.L.); andrea.kwa.l.h@sgh.com.sg (L.H.A.K.); 2Department of Pharmacy, Southwest Hospital of Army Medical University, Chongqing 400038, China; liufang0209@163.com

**Keywords:** co-amoxiclav, safety, efficacy, diabetic foot infection, *Enterobacterales*

## Abstract

**Background**: With increasing pharmacokinetic evidence suggesting the inadequacy of conventional dose intravenous co-amoxiclav (IVCA) 1.2 g Q8H in targeting *Enterobacterales*, our institution antibiotic guidelines optimised dosing recommendations for diabetic foot infection (DFI) management to 1.2 g Q6H in August 2023. In this study, we aim to evaluate the efficacy and safety of the optimised dose IVCA in DFI treatment. **Methods**: In this single-centre cohort study, patients ≥ 21 years with DFI, creatinine clearance ≥ 50 mL/min, and weight > 50 kg, who were prescribed IVCA 1.2 g Q8H (standard group (SG)), were compared with those prescribed IVCA 1.2 g Q6H (optimised group (OG)). Patients who were pregnant, immunocompromised, had nosocomial exposure in last 3 months, or received < 72 h of IVCA were excluded. The primary efficacy outcome was clinical deterioration at end of IVCA monotherapy. The secondary efficacy outcomes include 30-day readmission and mortality, empiric escalation of antibiotics, lower limb amputation, and length of hospitalisation. The safety outcomes include hepatotoxicity, renal toxicity, and diarrhoea. **Results:** There were 189 patients (94 in SG; 95 in OG) included. Patients in SG (31.9%) were twice as likely to experience clinical deterioration compared to OG (16.8%) (odds ratio: 2.31, 95% confidence interval: 1.16–4.62, *p* < 0.05). There were statistically more patients who had 30-day all-cause mortality in SG (5.3%) compared to OG (0%) (*p* < 0.05). Furthermore, 30-day readmission due to DFI in SG (26.6%) was higher compared to OG (11.6%) (*p* < 0.05). Empiric escalation of IV antibiotics was required for 14.9% patients in SG and 6.3% patients in OG (*p* = 0.06). There was no statistical difference for lower limb amputation (*p* = 0.72), length of hospitalisation (*p* = 0.13), and the occurrence of safety outcomes in both groups. **Conclusions:** This study suggests IVCA 1.2 g Q6H is associated with the decreased likelihood of clinical deterioration and is likely as safe as IVCA 1.2 g Q8H. The optimised dose of IVCA may help reduce the use of broad-spectrum antibiotics due to clinical deterioration.

## 1. Introduction

Diabetes mellitus (DM) is increasingly prevalent globally and is of growing concern due to its economic and social burden [1,2]. Globally, DM is the eighth leading cause of disease and mortality [1,3]. Currently, around 400,000 Singaporeans live with DM, and this number is projected to increase to 1 million by 2050 [4]. In 2019, DM was the sixth leading cause of burden of disease and mortality combined in Singapore [5].

Diabetic foot infection (DFI) is a devastating complication of DM, which develops from a diabetic foot ulcer via a complex interplay of biomechanical, neurological, and vascular pathways due to diabetic neuropathy and peripheral artery disease [6]. When left untreated, DFI is associated with substantial morbidity, with many patients receiving a lower limb amputation (LLA) that impairs physical function [6]. DFI leads to frequent healthcare provider visits, wound care, and increased healthcare costs, translating to a lower quality of life for this patient population [2,6].

The majority of DFIs are polymicrobial in nature [7,8,9]. Intravenous co-amoxiclav (IVCA) offers broad-spectrum coverage against Gram-positive organisms and Gram-negative organisms, both of which are commonly implicated in DFI [7,8,9]. Moreover, IVCA has good penetration to soft tissues and bones in the lower limbs [10]. Hence, IVCA is an excellent option for the management of DFI in patients from the community who do not have risk factors for nosocomial infections, such as the absence of prior antibiotic exposure in the past 3 months and recent hospitalisation [11].

At Singapore General Hospital (SGH), our in-house guidelines recommended IVCA at the conventional dose of 1.2 g Q8H to target the implicated organisms in DFI, which corroborates with the literature’s recommendations [7,12]. We do not initiate renal adjustments of IVCA across all renal functions in light of the concerns over the inadequacy of doses, and that IVCA is generally tolerable, with gastrointestinal symptoms being the most common side effect [13].

There is an increasing amount of evidence suggesting that the conventional dose of IVCA 1.2 g Q8H is inadequate in targeting *Enterobacterales* with amoxicillin (of the co-amoxiclav component), with a minimum inhibitory concentration of (MIC) > 8 µg/mL, based on European Committee on Antimicrobial Susceptibility Testing (EUCAST) [14]. Available pharmacokinetics/pharmacodynamics (PK/PD) data showed that IVCA 1.2 g Q6H achieved a probability of target attainment of 90% (for PK/PD target of >40% time above MIC) for *Enterobacterales* with an MIC of ≤4/2 μg/mL [15,16]. Hence, based on the current susceptible breakpoint of ≤8/4 µg/mL for co-amoxiclav towards *Enterobacterales*, the Clinical and Laboratory Standards Institute (CLSI) recommends IVCA 1.2 g Q6H [17]. In addition, EUCAST recommends IVCA 1.2 g Q6-8H (standard dose) or IVCA 1.2 g Q8H with IV amoxicillin 1 g Q8H (high-dose), without specifying the infection type [18]. Appropriate dosing of IVCA is crucial for DFI management, which has been associated with high rates of lower limb amputation (13.3 per 100,000 population in 2013) and 30-day mortality (11.1%) in Singapore [2]. Furthermore, patients with DFI tend to have compromised blood flow due to poor perfusion [2,6], which aggravates the concern of inadequate antibiotic penetration to infected lower limbs with the conventional dose of IVCA 1.2 g Q8H. At our institution, physicians are less likely to prescribe high-dose IVCA 1.2 g Q8H with IV amoxicillin 1 g Q8H upfront, due to lack of experience with such an aggressive dosing regimen. Therefore, from August 2023, the antimicrobial stewardship unit (ASU) team revised the in-house antibiotic guidelines and recommended an optimised dose of IVCA at 1.2 g Q6H for patients with actual body weight (ABW) > 50 kg and creatinine clearance (CrCl) ≥ 50 mL/min.

To our knowledge, whilst there have been studies pertaining to the use of high-dose co-amoxiclav in acute sinusitis and melioidosis [19,20], there is a lack of evidence for the efficacy of the optimised dose of IVCA in DFI. Although IVCA 1.2 g Q8H is generally tolerable in our local population, clinicians are hesitant to use the optimised dose of IVCA 1.2 g Q6H due to concerns over the higher risk of possible drug-induced hepatotoxicity and gastrointestinal adverse events, including *Clostridioides difficile* infection (CDI) [21,22,23]. Hence, in this study, we aim to evaluate the efficacy and safety of the optimised dose of IVCA in patients with DFI.

## 2. Results

### 2.1. Demographics and Clinical Characteristics of Patients

A total of 189 patients (94 in standard group (SG) and 95 in optimised group (OG)) were included in this study (Table 1). Of these, 138 of 189 patients (73.0%) were males, 52 of 189 patients (27.5%) were obese with body mass index ≥ 30 kg/m^2^, the median age was 61 years (interquartile range (IQR): 53–69 years), and 51 of 189 patients (27.0%) were cigarette smokers. Most patients (142 of 189 (75.1%)) had moderate (International Working Group on the Diabetic Foot/Infectious Diseases Society of America (IWGDF/IDSA) DFI grade 3) or severe DFI (IWGDF/IDSA DFI grade 4); 83 patients (43.9%) had deep-seated diabetic foot osteomyelitis (DFO). The median duration of IVCA therapy was 6 days (IQR: 5–9 days).

### 2.2. Microbiology

A total of 156 out of 189 (75 in SG; 81 in OG) patients (82.5%) had organisms isolated in their first intra-operative lower limb culture result. The most commonly isolated pathogens were *Staphylococcus* (79 out of 323 isolates, 24.5%) and *Streptococcus* (57 out of 323 isolates, 17.6%). Of the 323 isolates, 145 (81 in SG; 64 in OG) were Gram-positive, 153 (86 in SG; 67 in OG) were Gram-negative, 20 (8 in SG; 12 in OG) were anaerobes, and 5 (2 in SG; 3 in OG) were fungi.

Amongst the 153 Gram-negative isolates, 126 (82.4%) isolates (70 in SG; 56 in OG) were *Enterobacterales*. There were 33 (19 in SG; 14 in OG) *Enterobacterales* isolates (26.2%) with borderline susceptibility to amoxicillin (i.e., MIC = 4–8 µg/mL). Fifty-one (40.5%) *Enterobacterales* isolates (27 in SG; 24 in OG) were tested as non-susceptible to amoxicillin according to the CLSI breakpoint for susceptibility (MIC > 8 µg/mL).

A total of 45 out of 94 patients (47.9%) in SG and 49 out of 95 (51.6%) in OG had a culture-directed switch of antibiotics. There were patients in both arms who had bacteria (non-*Enterobacterales*) resistant to the co-amoxiclav isolated in their lower limb culture. All of these patients had a culture-directed switch of antibiotics and the discontinuation of IVCA therapy by their treating physician (i.e., switch of IVCA to IV vancomycin or PO clindamycin for methicillin-resistant *Staphylococcus aureus* (MRSA), switch of IVCA to IV piperacillin-tazobactam, IV ciprofloxacin/levofloxacin, or IV cefepime for *Pseudomonas aeruginosa*).

### 2.3. Primary Efficacy Outcome

#### 2.3.1. Overall Clinical Deterioration in Standard Group vs. Optimised Group

There was a significantly greater number of patients (30/94, 31.9%) in the SG who experienced clinical deterioration compared to patients (16/95, 16.8%) in the OG (odds ratio 2.31, 95% confidence interval (CI): 1.16–4.62, and *p* < 0.05) (Table 2).

#### 2.3.2. Clinical Deterioration Stratified by IWGDF/IDSA DFI Classification

Notably, amongst patients who had severe DFI (grade 4), there was a statistically significant higher number of patients who had clinical deterioration (10/18, 55.6%) in the SG compared to patients in the OG (7/28, 25.0%) (odds ratio 3.75, 95% CI: 1.06–13.27, and *p* < 0.05) (Table 3). This observation was similar among patients who had moderate DFI (grade 3), with a statistically significant higher number of patients who had clinical deterioration (20/51, 39.2%) in the SG compared to patients (9/45, 20.0%) in the OG (odds ratio 2.58, 95% CI: 1.03–6.49, and *p* < 0.05) (Table 3). None of the patients who had mild DFI (grade 2) experienced clinical deterioration.

#### 2.3.3. Clinical Deterioration Stratified by Presence of Sensitive Gram-Positive Organisms or Polymicrobial Growth of Organisms Isolated

Among patients with Gram-positive organisms (such as methicillin-sensitive *Staphylococcus aureus*, *Streptococcus* spp.) isolated that were tested to be sensitive to co-amoxiclav, there was a higher proportion of patients in the SG (18/43 (41.9%)) who had clinically deteriorated compared to the OG (8/53 (15.1%)) (*p* < 0.05) (Table 3). For patients with polymicrobial growth in their lower limb culture, fewer patients in the OG (12/55 (21.8%)) experienced clinical deterioration compared to patients in the SG (25/56 (44.6%)) (*p* < 0.05) (Table 3).

#### 2.3.4. Clinical Deterioration Stratified by MIC of Enterobacterales Isolated

Amongst those who had clinical deterioration, 23/30 patients in the SG (76.7%) had at least one *Enterobacterales* isolated in their first intra-operative lower limb culture result, as opposed to 10/16 patients (62.5%) in the OG (*p* = 0.31) (Table 2). For patients who had sensitive *Enterobacterales* with amoxicillin MIC ≤ 2 µg/mL isolate, fewer of them in the OG (2/15 (13.3%)) experienced clinical deterioration compared to those in the SG (7/21 (33.3%)) (*p* = 0.25) (Table 3). For those with *Enterobacterales* of borderline susceptibility towards amoxicillin (MIC 4–8 µg/mL), there were slightly more patients in the OG (4/14 (28.6%)) who experienced clinical deterioration, compared to those in the SG (4/19 (21.1%)) (*p* = 0.70) (Table 3). For patients with resistant *Enterobacterales* isolate (amoxicillin MIC > 8 µg/mL), a statistically significant fewer number of patients in the OG experienced clinical deterioration compared to those in the SG (6/23 (26.1%) in OG vs. 20/27 in SG (74.1%), *p* < 0.05) (Table 3).

### 2.4. Secondary Efficacy Outcomes

There was a statistically higher proportion of patients who had 30-day all-cause mortality from the start of IVCA therapy in the SG (5/94, 5.3%) compared to the OG (0/95, 0%) (*p* < 0.05). Furthermore, 30-day readmission due to DFI in the SG (25/94, 26.6%) was higher compared to the OG (11/95, 11.6%) (*p* < 0.05) (Table 2). Although not statistically different, the empiric escalation of IVCA to a broader spectrum antibiotic for DFI management was notably higher in the SG (14/94, 14.9%) than the OG (6/95, 6.3%) (*p* = 0.06) (Table 2).

There was no difference in the length of hospitalisation or number of LLA in both the SG and OG.

### 2.5. Safety Outcomes

Non-fatal adverse events occurred in 9/94 (9.6%) patients in the SG compared to 11/95 (11.6%) patients in the OG (*p* = 0.65) (Table 4).

Hepatotoxicity due to IVCA therapy occurred in two patients (2.1%) and one patient (1.1%) in the SG and OG, respectively (*p* = 0.62). One patient in the SG had a raised alanine transaminase (ALT) with bilirubin elevation (Naranjo score +5 [probable]). The remaining two patients in the SG and OG had raised alkaline phosphatase (ALP) with accompanying gamma-glutamyl transferase (GGT) elevation (Naranjo score +5 [probable] and +3 [possible], respectively).

A total of 7 of 94 patients (7.4%) in the SG developed diarrhoea compared with 10 of 95 patients (10.5%) in the OG (*p* = 0.46). Notably, none of the patients in the OG developed CDI whereas amongst the seven patients in the SG, two developed CDI (diarrhoea onset on Day 7 and 10 of IVCA treatment, respectively; Naranjo score +7 [probable] for both patients). Both patients were treated with an adequate course of 10 and 14 days of PO vancomycin 125 mg Q6H, respectively.

None of the patients in both groups developed renal toxicity (acute kidney injury).

## 3. Discussion

### 3.1. Results Interpretation and Implications

To our knowledge, this is the first study that evaluated the safety and efficacy of an optimised dose of IVCA in patients with DFI. In our study, we showed that patients who received a conventional dose of IVCA 1.2 g Q8H were twice as likely to experience clinical deterioration (30/94 patients (31.9%)) compared to those who received the optimised dose of IVCA 1.2 g Q6H (16/95 patients (16.8%)) (odds ratio 2.31, 95% CI: 1.16–4.62, and *p* < 0.05). Notably, patients who received the conventional dose of IVCA were at least 2.5 to 3.75 times more likely to experience clinical deterioration than the optimised dose of IVCA in patients who had moderate DFI (IWGDF/IDSA grade 3) (odds ratio 2.58, 95% CI: 1.03–6.49, and *p* < 0.05) and severe DFI (IWGDF/IDSA grade 4) (odds ratio 3.75, 95% CI: 1.06–13.27, and *p* < 0.05). This suggests that there is a benefit in using the optimised dose of IVCA in patients with DFI, especially in patients with extensive infection or who are showing signs of systemic toxicity and metabolic instability. Additionally, we demonstrated good patient outcomes, with the OG arm showing a lower incidence of 30-day all-cause mortality and 30-day readmission due to DFI.

The rationale of initiating the optimised dose of IVCA was mainly to target *Enterobacterales* with borderline amoxicillin susceptibility (MIC = 4–8 µg/mL). Nonetheless, our study findings showed that this dosing is also beneficial in patients who are infected with Gram-positive organisms. Amongst patients who had Gram-positive organisms or polymicrobial organisms isolated, there were fewer patients who clinically deteriorated in the OG compared to the SG. Furthermore, amongst patients who have *Enterobacterales* isolate with amoxicillin MIC ≤ 2 µg/mL, there appears to be a smaller proportion of patients in the OG who experienced clinical deterioration compared to those in the SG (2/15 (13.3%) in OG vs. 7/21 (33.3%) in SG, *p* = 0.25). This suggests that even in patients with DFI who have a sensitive *Enterobacterales* with very low amoxicillin MIC of ≤2 µg/mL, they may still benefit from the receipt of the optimised dose of IVCA 1.2 g Q6H as compared to the conventional dose of IVCA 1.2 g Q8H.

Notably, a significant 40.5% of *Enterobacterales* isolates in this study had amoxicillin MIC > 8 µg/mL. Amongst patients with ≥1 *Enterobacterale*(s) isolated from their first intra-operative lower limb culture with amoxicillin MIC > 8 µg/mL, there were fewer patients who experienced clinical deterioration in the OG (6/23, 26.1%) compared to the SG (20/27, 74.1%) (*p* < 0.05). While use of IVCA at the conventional dose of 1.2 g Q8H for *Enterobacterales* with amoxicillin MIC > 8 µg/mL has been reported to be predictive of therapeutic failure [14,16,24], our study suggests that an empiric optimised IVCA dose of 1.2 g Q6H reduces the risk of clinical deterioration. On the other hand, there appears to be a slightly greater proportion of patients with clinical deterioration in the OG amongst those with *Enterobacterales* isolate of borderline susceptibility (amoxicillin MIC = 4–8 µg/mL) (4/14 (28.6%) in OG vs. 4/19 in SG (21.1%), *p* = 0.70). As DFI is predominantly a polymicrobial infection [7,8,9], we postulate that these discrepancies could be attributed to the presence of colonisers in the relevant intra-operative lower limb culture. Determining the significance of organisms isolated from lower limb cultures remains a challenge in the management of DFI; isolated organisms could either be colonisers or true pathogens. This challenge in differentiation is even more apparent in cases where patients refused source control due to sociocultural issues [25], as only bedside swab cultures could be obtained. This highlights the importance of our vascular surgeons and infectious diseases physicians in performing a physical bedside assessment of the wound status to better assess clinical progress and ascertain the need to treat all organisms isolated from intra-operative lower limb cultures.

Amongst the four patients in the OG who experienced clinical deterioration and had *Enterobacterales* with amoxicillin MIC 4–8 µg/mL isolated from their intra-operative lower limb culture, all had DFO, IWGDF/IDSA DFI grade 3, and CrCl > 100 mL/min. This suggests that the optimised dose of IVCA 1.2 g Q6H might still be insufficient to achieve free co-amoxiclav levels above MIC for at least 40% of the dosing interval, an established PK/PD efficacy metric for β-lactams [16]. Perhaps, an even higher dose of amoxicillin (IVCA 1.2 g Q8H with IV amoxicillin 1 g Q8H), corresponding to a daily intake of 6 g of amoxicillin, may be required as per the high dose recommendation by EUCAST [18] in patients with deep-seated infections and good renal function. However, the safety and tolerability of such high doses will be a concern among clinicians. There have been studies to evaluate the safety and efficacy of amoxicillin based on therapeutic drug monitoring (TDM) in a non-critically ill population [26]. While a correlation between amoxicillin levels and clinical improvement cannot be established, an amoxicillin level of >40 µg/mL was predictive of the occurrence of adverse events [27]. In our study, the optimised dose of IVCA 1.2 g Q6H appears tolerable. Yet, safety data for doses higher than 1.2 g Q6H is lacking and perhaps TDM should still be advocated in stable patients who require an even higher IVCA dose. On the other hand, in critically ill patients with variable and unpredictable PK profiles, there will certainly be a utility in performing TDM based on extrapolation from existing β-lactam TDM studies [26], to ensure the adequacy of doses even when the optimised dose of IVCA is used.

Incidence of severe adverse events due to IVCA is generally low, with hepatotoxicity occurring in 1 in every 2350 patients prescribed with co-amoxiclav [21]. Although our study was not powered to detect subtle differences in the incidence of adverse events, we could possibly conclude that the use of IVCA 1.2 g Q6H remains generally safe compared to IVCA 1.2 g Q8H. Diarrhoea, which occurred in about 10% of patients in the OG, was comparable to the incidence rate of other commonly prescribed antibiotics [28]. None of the patients in both groups developed other types of toxicity (including acute kidney injury) attributed to IVCA. Balancing the benefits and risks of clinical deterioration, readmission, and mortality, IVCA 1.2 g Q6H would still be preferred over 1.2 g Q8H.

Management of DFI goes beyond antimicrobial therapy; re-establishing vascular supply to the limbs coupled with prompt surgical source control is vital. The overall number of lower limb revascularisation (LLR) was similar amongst patients who received the optimised dose of IVCA (40/95 (42.1%)) and the conventional dose of IVCA (46/94 (48.9%)). This suggests that patients in both groups had similar lower limb vascularity upon admission to the emergency department. The overall number of LLA was similar amongst patients who received the optimised dose of IVCA (44/95 (46.8%)) and the conventional dose of IVCA (42/94 (44.2%)) (*p* = 0.72). The overall number of source control (LLR and/or LLA) performed was similar amongst these two groups of patients (79/95 (83.2%) patients in OG vs. 76/94 (80.9%) patients in SG). The high percentage of source control performed in both groups is a testament that LLR and LLA remain the cornerstone for DFI management [7]; the prescription of appropriate IV antibiotics serve as an adjunct to its management to lessen bacterial burden and reduce the risk of infective complications [7].

### 3.2. Antimicrobial Stewardship

From the antimicrobial stewardship perspective, it is important to encourage physicians to be compliant with our in-house guidelines and dose IVCA adequately. In our local context, our physicians are conservative and have a low threshold to escalate antimicrobial coverage from IVCA to another agent with pseudomonal coverage (e.g., piperacillin-tazobactam) even though the patient has yet to receive an adequate exposure of IVCA ≥ 72 h. The prescription of broad-spectrum antimicrobials may drive antimicrobial resistance, adverse events (including CDI), mortality, and morbidity [29]. In our study, though not statistically significant, we have demonstrated that a lower proportion of patients who were in the OG (6/95, 6.3%) had IVCA empirically escalated compared to patients in the SG (14/94, 15.9%) (*p* = 0.06), suggesting that perhaps patients demonstrated quicker recovery with the OG and hence were less likely to require escalation in antibiotic therapy from the physicians’ perspective.

Additionally, there is increasing evidence to suggest that an optimised β-lactam dosing regimen maximises bacterial cell kill and minimises risk for β-lactam resistance due to subtherapeutic levels [30,31,32]. While antibiotic resistance was not evaluated in our study, we believe that optimising IVCA dosing is an antimicrobial stewardship strategy to slow down the emergence of resistance in DFI management.

### 3.3. Limitations

Our study is not without its limitations. Firstly, due to logistical challenges that involve obtaining patient consent and scheduling LLR in the procedural rooms, there were unavoidable delays in performing LLR. Timing to LLR has been associated with therapeutic failures in ischemic DFI [33] and, unfortunately, it is unknown regarding the extent to which delays in LLR have contributed to clinical deterioration, 30-day readmission, and mortality due to DFI in our study.

Secondly, it was impossible to ascertain if complete source control was achieved through surgical debridement for patients with DFI. It was challenging to correctly estimate whether intra-operatively clear surgical margins have been achieved when infected bones were resected or when infected soft tissues were debrided or amputated away. This could have influenced clinical deterioration and 30-day mortality if imbalances were present.

Thirdly, this study was not conducted in the same time period for both the SG and OG. Nonetheless, we observed that there were a higher proportion of patients in the OG with IWGDF/IDSA DFI grade 4 compared to the SG (28/95 patients in the OG (29.5%) vs. 18/94 patients in the SG (19.1%)). Additionally, patients in the OG were significantly younger (median 60 vs. 65 years old, *p* < 0.05) and had a numerically higher median HBA1c (median 8.9% vs. 7.8%, *p* = 0.09) compared to the SG (Table 1). A younger age with higher HbA1c has been associated with an increased likelihood of DFI recurrence and hospitalisation [34]. Hence, despite having a less favourable patient population in the OG, our study was still able to prove that patients receiving IVCA 1.2 g Q6H in the OG have better clinical outcomes than the SG.

Lastly, the extent of wound recovery and infection management is also dependent on the patients’ compliance with their chronic medications and their competency at wound dressing post-surgery. Chronic medications that affect atherosclerotic risk (blood thinners, anticoagulants, and statins), venous flow (diosmin and pentoxifylline), and blood sugar control (antidiabetic agents) can affect wound healing. Moreover, patients and caregivers may not be trained to the same competency level in wound dressing, and there may be use of different agents for wound dressing, including iodosorb, povidone-iodine, silver nitrate, and chlorhexidine. We acknowledge that the management of DFI requires a multi-disciplinary approach beyond optimising antibiotic treatment: (a) a vascular surgeon to perform adequate source control (in the form of LLR, lower limb surgical debridement, and/or LLA); (b) a pharmacist to motivate patients on compliance to their chronic medications; and (c) a nurse/podiatrist to manage the DFI wound adequately [3,7,35].

## 4. Materials and Methods

### 4.1. Study Design

In this single-centre cohort study, we included patients admitted for inpatient stay in SGH (tertiary care hospital), who fulfilled all the following criteria: (a) age ≥ 21 years old; (b) diagnosed with limb-threatening DFI on presentation; (c) DFI occurred in the community (i.e., antibiotic-naïve and no recent hospitalisation within the last 3 months); (d) CrCl ≥ 50 mL/min; and (e) ABW ≥ 50 kg.

Patients were excluded if they fulfilled any of the following criteria: (a) labelled with allergy to β-lactam; (b) development of acute kidney injury or CrCl dipped below 50 mL/min during IVCA course; (c) expected mortality within 48 h; (d) immunocompromised host; (e) pregnant; (f) has concomitant infection aside from DFI; (g) received IVCA for duration ≤72 h; and (h) receipt of concomitant antibiotic in addition to IVCA for current admission.

All patients were prescribed IVCA (Trademark: Curam^®^ (Sandoz GmbH, Kundl, Austria)) [12], available in a 5:1 ratio with amoxicillin 1000 mg and clavulanic acid 200 mg per vial. Our patients received either conventional dose IVCA 1.2 g Q8H from June 2022 to July 2023 (SG), or optimised dose IVCA 1.2 g Q6H from August 2023 to September 2024 (OG). All patients were stratified based on the IWGDF/IDSA classification on the severity of DFI (graded as a score from 1 to 4, with a subclass for osteomyelitis) (Appendix A) [7]. Patients were deemed to have a clinical diagnosis of DFO if they fulfilled ≥1 positive finding(s) from our institution’s routine workup for DFO, including: (i) radiological findings compatible with osteomyelitis (plain X-ray and/or magnetic resonance imaging of the foot); (ii) positive probe to bone test; (iii) intra-operative findings consistent with diagnosis of DFO; and (iv) microbial growth in proximal bone chip cultures [7].

### 4.2. Microbiology

Only microorganisms isolated from the first culture performed for the DFI management were taken into consideration for this study. Susceptibility of isolated bacteria to co-amoxiclav was performed by the institution’s microbiology laboratory according to methodology by CLSI. *Enterobacterales* MICs for co-amoxiclav were determined by VITEK^®^ 2 system, with MIC ≤ 8/4 µg/mL considered as susceptible as per CLSI recommendation [17].

### 4.3. Data Collection

Patient demographics (age, sex, weight, BMI, smoking status, and HbA1c), antibiotic allergy history, renal function, antibiotic-related information (dosing regimen, duration of IVCA therapy, adverse events including hepatotoxicity, and diarrhoea), microorganisms isolated in the first intra-operative lower limb culture, and their respective MIC (for *Enterobacterales*) or susceptibility panel (for all other microorganisms) were collected using a standardised data collection form.

Patient data were collected retrospectively and prospectively, for patients in SG and OG, respectively. All data was collected solely from the patient’s electronic medical records. This study was exempted from our institutional ethics board review (Singhealth Institutional Review Board Reference 2022/2560).

### 4.4. Primary Efficacy Outcome

The primary efficacy outcome was clinical deterioration during the IVCA course and was determined upon the discontinuation of IVCA monotherapy. For patients with concomitant antibiotics added during IVCA therapy (i.e., IV vancomycin or PO clindamycin for MRSA isolated from their first intra-operative lower limb culture), the patient will be assessed for clinical deterioration prior to the addition of the anti-MRSA agent. Patients were deemed to have experienced clinical deterioration if they have a worsening wound condition, in combination with either signs that were suggestive of worsening sepsis as defined by systemic inflammatory response syndrome (SIRS) criteria [7,36] or worsening of inflammatory markers or both (Appendix A).

### 4.5. Secondary Efficacy Outcomes

The secondary efficacy outcomes include: (a) all-cause 30-day readmission (from day of discharge) and 30-day readmission due to DFI; (b) all-cause 30-day mortality (from start of IVCA therapy) and 30-day mortality due to DFI; (c) length of hospitalisation due to DFI in days; (d) number of patients who underwent LLA; and (e) number of empiric escalation of IVCA to a broader-spectrum antibiotic.

### 4.6. Safety Outcomes

Drug-induced hepatotoxicity was based on the definition by the European Association for the Study of the Liver [37]. Patients who developed either of the following were deemed to have developed hepatotoxicity within 30 days of IVCA initiation: (a) ≥5× upper limit of normal (ULN) elevation in ALT; (b) ≥2× ULN elevation in ALP with accompanying elevation in GGT and absence of known bone pathology that will confound rise in ALP; (c) ≥3× ULN elevation in ALT and a ≥2× ULN simultaneous elevation of total bilirubin concentration.

Diarrhoea was defined as the passage of ≥3 loose or liquid stools per day, equivalent to Type 6–7 stool as per the Bristol Stool Chart while receiving IVCA therapy [23,38]. Patients with diarrhoea due to laxative usage were excluded from the count. In our institution, *C. difficile* workup was performed for patients with unexplained and new onset diarrhoea (≥3 Type 7 stools in 24 h). *C. difficile* is considered as positive when there is either (1) the simultaneous detection of the *C. difficile* glutamate dehydrogenase (GDH) antigen and toxins A and B based on rapid membrane enzyme immunoassay (C DIFF QUIK CHEK COMPLETE^®^, TechLab) or (2) the presence of Toxin B gene of toxigenic *Clostridioides* (tcdB gene) from a multiplex real-time polymerase chain reaction (PCR) using GeneXpert^®^ *C. difficile* (Cepheid). Oral vancomycin was administered for all patients with a definitive diagnosis of *C. difficile* infection [39].

Renal toxicity, in the form of acute kidney injury, was defined based on the definition by the Kidney Disease: Improving Global Outcomes (KDIGO). Patients who developed either of the following were deemed to have acute kidney injury within 30 days of IVCA initiation: (a) rise in serum creatinine by at least 50% from baseline; (b) absolute increase in serum creatinine by ≥0.3 mg/dL; and (c) urine output < 0.5 mL/kg/h for ≥6 consecutive hours [40].

The Naranjo Adverse Drug Reaction Probability Scale was used to determine the likelihood of adverse events associated with the use of IVCA therapy [41].

### 4.7. Statistical Analysis

We performed all statistical analyses using IBM^®^ SPSS^®^ Version 26.0 for Windows (IBM Corporation, Armonk, New York, UY, USA, 2019). The Chi-Square test/Fisher’s exact test (as appropriate) was used for categorical variables, with the odds ratio and two-sided 95% CI calculated for a statistically significant result set at p = 0.05. The Mann–Whitney U test was used for continuous variables (not normally distributed). Descriptive statistics were used to summarise data and were tabulated as median (IQR) or n (%).

## 5. Conclusions

The optimised dose of IVCA 1.2 g Q6H is associated with the reduced likelihood of clinical deterioration and is likely as safe as the conventional dose of IVCA 1.2 g Q8H in the treatment of DFI. The optimised dose of IVCA 1.2 g Q6H may help reduce use of even broader spectrum antibiotics beyond co-amoxiclav via antibiotic escalation due to clinical deterioration. Further studies on higher doses of amoxicillin (IVCA 1.2 g Q8H with IV amoxicillin 1 g Q8H) for patients who have deep-seated DFI and augmented renal clearance, or IVCA 1.2 g Q6H for patients with deep-seated DFI and suboptimal renal function of CrCl < 50 mL/min) are required to determine if they are safe and effective for this patient population.

## Figures and Tables

**Table 1 antibiotics-14-00758-t001:** Sociodemographic and Clinical Characteristics of Patients.

Baseline Characteristics	Standard Group IVCA 1.2 g Q8H (*n* = 94)	Optimised Group IVCA 1.2 g Q6H (*n* = 95)	*p*-Value
**Age (years)**	**65 (54–70)**	**60 (52–64)**	**<0.05**
**Male**	69 (73.4%)	69 (72.6%)	0.91
**Actual body weight (kg)**	75 (67–87)	79 (71–88)	0.13
**BMI range (kg/m^2^)**			
≤27.4	54 (57.5%)	48 (50.5%)	0.34
27.5–29.9	13 (13.8%)	22 (23.2%)	0.10
≥30	27 (28.7%)	25 (26.3%)	0.71
**Smoker**	24 (25.5%)	27 (28.4%)	0.66
**HbA1c within last 3 months (%)**	7.8 (7–9.2)	8.9 (7–10.3)	0.09
**IWGDF/IDSA DFI classification**			
2	25 (26.6%)	22 (23.2%)	0.59
3	22 (23.4%)	14 (14.7%)	0.13
3 (Osteomyelitis)	29 (30.9%)	31 (32.6%)	0.79
4	8 (8.5%)	15 (15.8%)	0.13
4 (Osteomyelitis)	10 (10.6%)	13 (13.7%)	0.52
**≥1 organism(s) isolated in initial cultures**	75 (79.8%)	81 (85.3%)	0.32
**≥1 Gram (+) organisms** ¥	56 (58.9%)	65 (68.4%)	0.21
***Arcanobacterium***	**0 (0%)**	**6 (6.3%)**	**<0.05**
Sensitive to co-amoxiclav	0 (0%)	2 (2.1%)	0.50
Intermediate to co-amoxiclav	0 (0%)	4 (4.2%)	0.12
***Corynebacterium***	6 (6.4%)	6 (6.3%)	0.99
Intermediate to co-amoxiclav	5 (5.3%)	6 (6.3%)	0.77
Resistant to co-amoxiclav	1 (1.1%)	0 (0%)	0.50
***Enterococcus***	19 (20.2%)	10 (10.5%)	0.07
Resistant to co-amoxiclav	3 (3.2%)	3 (3.2%)	>0.99
**≥1 *Staphylococcus*** ¥	33 (35.1%)	40 (42.1%)	0.32
CoNS	8 (8.5%)	7 (11.8%)	0.77
Resistant to co-amoxiclav	0 (0%)	1 (1.1%)	>0.99
MRSA	5 (5.3%)	6 (9.8%)	0.77
MSSA	22 (23.4%)	30 (31.6%)	0.21
***Streptococcus***	**21 (22.3%)**	**36 (37.9%)**	**<0.05**
**Other Gram (+) organisms ***	0 (0%)	3 (3.2%)	0.25
**≥1 Gram (−) organisms** ¥	60 (63.8%)	57 (60%)	0.59
**≥1 *Enterobacterales* **** ¥	51 (54.3%)	46 (48.4%)	0.42
MIC ≤ 2	21 (22.3%)	15 (16.0%)	0.25
MIC 4–8	19 (20.2%)	14 (14.7%)	0.32
MIC > 8	27 (28.7%)	24 (25.3%)	0.59
Unknown MIC	3 (3.2%)	3 (3.2%)	0.99
**Non-fermenters**	16 (17.0%)	11 (11.6%)	0.29
*Pseudomonas*	14 (14.9%)	11 (11.6%)	0.50
*Stenotrophomonas*	2 (2.1%)	0 (0%)	0.25
**Anaerobes**	8 (8.5%)	12 (12.6%)	0.36
**Fungi**	2 (2.1%)	3 (3.2%)	>0.99
**Duration of IVCA therapy (days)**	6 (5–8)	6 (5–9)	0.76
**Underwent lower limb debridement**	56 (59.6%)	49 (51.6%)	0.27

Data are expressed as a number (percentage) for categorical variables and median (interquartile range) for continuous variables. The *p*-values less than 0.05 are bolded. **NOTE:** None of the *Enterococcus* isolates were vancomycin resistant. All MSSA, *Streptococcus*, and anaerobes isolates were susceptible to co-amoxiclav. ¥ Summation of organisms may not tally with total count, as some cultures have polymicrobial infection by ≥2 different organisms from the same category (category refers to Gram-positive organisms, *Staphylococcus*, Gram-negative organisms and *Enterobacterales*). * Other Gram-positive organisms isolated include those from the following genera: *Actinomyces*, *Gemella*, and *Solobacterium*. ** *Enterobacterales* in our study include organisms from the following genera: *Aeromonas*, *Citrobacter*, *Enterobacter*, *Escherichia*, *Klebsiella*, *Morganella*, *Proteus*, *Providentia*, and *Serratia*. MIC stated reflects the amoxicillin component of co-amoxiclav. **Abbreviations:** *BMI* Body Mass Index, *CoNS* Coagulase-Negative *Staphylococcus*, *DFI* Diabetic Foot Infection, *Gram* (−) Gram-negative, *Gram* (+) Gram-positive, *HbA1c* Glycated Haemoglobin, *IDSA* Infectious Diseases Society of America, *IVCA* Intravenous Co-Amoxiclav, *IWGDF* International Working Group on the Diabetic Foot, *MIC* Minimum Inhibitory Concentration, *MRSA* Methicillin-Resistant *Staphylococcus aureus*, and *MSSA* Methicillin-Sensitive *Staphylococcus aureus*.

**Table 2 antibiotics-14-00758-t002:** Efficacy outcomes of IVCA therapy for patients with DFI.

Efficacy Outcomes	Standard Group IVCA 1.2 g Q8H (*n* = 94)	Optimised Group IVCA 1.2 g Q6H (*n* = 95)	*p*-Value
**Clinical response at end of IVCA therapy**			
Clinical deterioration	**30 (31.9%)**	**16 (16.8%)**	**<0.05**
≥1 *Enterobacterale*(s) isolated from culture	23 (76.7%)	10 (62.5%)	0.31
**Empiric escalation of IVCA therapy**	14 (14.9%)	6 (6.3%)	0.06
**30-day readmission (from day of discharge)**	28 (29.8%)	18 (18.9%)	0.08
Readmission due to DFI	**25 (26.6%)**	**11 (11.6%)**	**<0.05**
**30-day mortality (from start of IVCA therapy)**	**5 (5.3%)**	**0 (0%)**	**<0.05**
Mortality due to DFI	1 (1.1%)	0 (0%)	0.50
**Length of hospitalisation due to DFI (days)**	11 (6–19)	9 (5–15)	0.13
**Lower limb amputations performed**	44 (46.8%)	42 (44.2%)	0.72

Data are expressed as a number (percentage) for categorical variables and median (interquartile range) for continuous variables. The *p*-values less than 0.05 are bolded. **Abbreviations:** *DFI* Diabetic Foot Infection, *IVCA* Intravenous Co-Amoxiclav.

**Table 3 antibiotics-14-00758-t003:** Clinical outcomes of patients after completion of IVCA therapy, stratified based on IWGDF/IDSA DFI classification and type of bacteria isolated.

	Standard Group IVCA 1.2 g Q8H (*n* = 94)	Optimised Group IVCA 1.2 g Q6H (*n* = 95)	*p*-Value
**IWGDF/IDSA DFI classification**			
**2 (Mild)**	25 (26.6%)	22 (23.2%)	0.30
Clinical improvement	25 (100%)	22 (100%)	>0.99
**3 (Moderate)**	51 (54.2%)	45 (47.4%)	0.34
Clinical deterioration	**20 (39.2%)**	**9 (20.0%)**	**<0.05**
Clinical improvement	**31 (60.8%)**	**36 (80.0%)**	**<0.05**
**4 (Severe)**	18 (19.1%)	28 (29.5%)	0.10
Clinical deterioration	**10 (55.6%)**	**7 (25.0%)**	**<0.05**
Clinical improvement	**8 (44.4%)**	**21 (75.0%)**	**<0.05**
**≥1 Gram (+) organism(s) isolated from culture**	56 (58.9%)	65 (68.4%)	0.21
**≥1 sensitive Gram (+) organism(s) to IVCA ***	43 (45.7%)	53 (55.8%)	0.17
Clinical deterioration	**18 (41.9%)**	**8 (15.1%)**	**<0.05**
Clinical improvement	**25 (58.1%)**	**45 (84.9%)**	**<0.05**
**≥1 *Enterobacterale*(s) isolated from culture** **	51 (54.3%)	49 (51.6%)	0.71
**MIC ≤ 2 µg/mL**	21 (22.3%)	15 (16.0%)	0.25
Clinical deterioration	7 (33.3%)	2 (13.3%)	0.25
Clinical improvement	14 (66.7%)	13 (86.7%)	0.17
**MIC 4–8 µg/mL**	19 (20.2%)	14 (14.7%)	0.32
Clinical deterioration	4 (21.1%)	4 (28.6%)	0.70
Clinical improvement	15 (78.9%)	10 (71.4%)	0.62
**MIC > 8 µg/mL**	27 (28.7%)	23 (24.2%)	0.48
Clinical deterioration	**20 (74.1%)**	**6 (26.1%)**	**<0.05**
Clinical improvement	**7 (25.9%)**	**17 (73.9%)**	**<0.05**
**Unknown MIC**	3 (3.2%)	3 (3.2%)	>0.99
Clinical deterioration	2 (66.7%)	1 (33.3%)	>0.99
Clinical improvement	1 (33.3%)	2 (66.7%)	>0.99
**Polymicrobial organisms isolated from culture**	56 (59.6%)	55 (57.9%)	0.82
Clinical deterioration	**25 (44.6%)**	**12 (21.8%)**	**<0.05**
Clinical improvement	**31 (55.4%)**	**43 (78.2%)**	**<0.05**

**NOTE:** Clinical deterioration and improvement were calculated based on each sub-group. Using patients in standard group with ≥1 *Enterobacterale*(s) isolated in their first lower limb culture and ≥1 *Enterobacterale*(s) with MIC ≤ 2 µg/mL (21 out of 94 patients, 22.3%) as an example, there were 7 out of 21 patients (33.3%) who experienced clinical deterioration. Data are expressed as a number (percentage) for categorical variables. The *p*-values less than 0.05 are bolded. * Denotes the number of patients with ≥1 Gram (+) organisms isolated from their first lower limb bedside swab/debridement culture (if patient did not receive lower limb amputation) or their first intra-operative lower limb culture (if patient received lower limb amputation) since admission for inpatient stay, and all Gram (+) organisms have demonstrated sensitivity to co-amoxiclav. ** Denotes the number of patients with ≥1 *Enterobacterales* isolated from their first lower limb culture since admission for inpatient stay. *Enterobacterales* in our study include organisms from the following genera: *Aeromonas*, *Citrobacter*, *Enterobacter*, *Escherichia*, *Klebsiella*, *Morganella*, *Proteus*, *Providentia*, and *Serratia*. MIC stated reflects the amoxicillin component of co-amoxiclav. **Abbreviations:** *DFI* Diabetic Foot Infection, *Gram* (+) Gram-positive, *IDSA* Infectious Diseases Society of America, *IVCA* Intravenous Co-Amoxiclav, *IWGDF* International Working Group on the Diabetic Foot, and *MIC* Minimum Inhibitory Concentration.

**Table 4 antibiotics-14-00758-t004:** Safety outcomes of IV co-amoxiclav for patients with DFI.

Safety Outcomes	Standard Group IVCA 1.2 g Q8H (*n* = 94)	Optimised Group IVCA 1.2 g Q6H (*n* = 95)	*p*-Value
**Hepatotoxicity**	2 (2.1%)	1 (1.1%)	0.62
**Diarrhoea**	7 (7.4%)	10 (10.5%)	0.46
*Clostridioides difficile* infection present	2 (28.6%)	0 (0%)	0.15
**Renal Toxicity**	0 (0%)	0 (0%)	>0.99

Data are expressed as a number (percentage) for categorical variables. **Abbreviations:** *DFI* Diabetic Foot Infection, *IV* Intravenous.

## Data Availability

The raw data supporting the conclusions of this article will be made available by the authors on request.

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
