# Peer review of "Optimising Regimen of Co-Amoxiclav (ORCA)—The Safety and Efficacy of Intravenous Co-Amoxiclav at Higher Dosing Frequency in Patients with Diabetic Foot Infection"

_antibiotics, 2025, doi:10.3390/antibiotics14080758_

Round 1

Reviewer 1 Report

Comments and Suggestions for Authors

The study is well-designed as a cohort comparison, providing  valuable and practical insights into the impact of a higher IVCA dosing frequency (1.2g Q6H) on clinical outcomes and safety in DFI (diabetic foot infection)patients.

Major points:

*Cohort Study Limitations: While acknowledged as a single-center cohort study, the retrospective nature of the Standard Group (SG) and the prospective nature of the Optimized Group (OG) [lines 128-130] introduce potential biases. Changes in clinical practice, diagnostic approaches, or patient characteristics over time (June 2022-July 2023 vs. August 2023-September 2024) could influence outcomes independently of the dosing regimen. This inherent limitation should be more explicitly discussed in the "Limitations" section.

*Microbiological Data and Breakpoints:

-MIC Breakpoint Clarity: The methods state that "Enterobacterales MICs for co-amoxiclav were determined by VITEK® 2 system, with MIC ≤ 8/4µg/mL considered as susceptible" [lines 119-120]. Please clarify if this breakpoint is specific to SGH or if it aligns with a recognized standard (e.g., CLSI). The provided research notes do not explicitly confirm this specific breakpoint for co-amoxiclav against Enterobacterales in CLSI/EUCAST.

-Significance of Non-Susceptible Isolates: The finding that 40.5% of Enterobacterales isolates were non-susceptible to amoxicillin (MIC > 8µg/mL) [lines 220-222] is a crucial piece of information. The discussion could further elaborate on how this high proportion of non-susceptible isolates might have influenced the observed clinical deterioration rates in both groups and reinforced the need for optimized dosing.

Specific Comments:

-Introduction :

The phrasing regarding EUCAST and CLSI recommendations for the 1.2g every 6 hours dosing regimen (lines 69–71) should be revisited to ensure precision and alignment with the latest published guidelines.

-Materials and Methods :

-Clarify the source/standard for the MIC breakpoint of ≤ 8/4µg/mL for Enterobacterales [lines 119-120].

-Discussion :

- Refine the statement about "even higher dose" and EUCAST recommendations [lines 363-364] to ensure accuracy with published guidelines. - While the safety data showed no significant difference, consider discussing if the study was powered to detect subtle increases in the incidence or severity of common adverse events like diarrhea, given that previous literature suggests higher doses can increase severity.  

Author Response

Please see the attachment for word document of author's reply to the review report (Reviewer 1)

Major points:

*Cohort Study Limitations: While acknowledged as a single-center cohort study, the retrospective nature of the Standard Group (SG) and the prospective nature of the Optimized Group (OG) [lines 128-130] introduce potential biases. Changes in clinical practice, diagnostic approaches, or patient characteristics over time (June 2022-July 2023 vs. August 2023-September 2024) could influence outcomes independently of the dosing regimen. This inherent limitation should be more explicitly discussed in the "Limitations" section.

Thank you for your comments. We agree that the retrospective nature of SG and prospective nature of OG may introduce potential biases. We have included this concern into the 3rd paragraph of Section 4.3 Limitations as follows (lines 452-460):

“Thirdly, this study was not conducted in the same time period for both SG and OG. Nonetheless, we observed that there were higher proportion of patients in OG with IWGDF/IDSA DFI grade 4 DFI compared to SG (28/95 patients (29.5%) vs. 18/94 (19.1%)). Additionally, patients in OG are significantly younger (median 60 vs. 65 years old, p<0.05) and had numerically higher median HBA1c (median 8.9% vs. 7.8%, p=0.09) compared to SG (Table 1). Younger age with higher HbA1c has been associated with increased likelihood of DFI recurrence and hospitalisation [40]. Hence, despite having a less favourable patient population in OG, our study was still able to prove that patients receiving IVCA 1.2g Q6H in OG have better clinical outcomes than SG.

Microbiological Data and Breakpoints

- MIC Breakpoint Clarity: The methods state that "Enterobacterales MICs for co-amoxiclav were determined by VITEK® 2 system, with MIC ≤ 8/4µg/mL considered as susceptible" [lines 119-120]. Please clarify if this breakpoint is specific to SGH or if it aligns with a recognized standard (e.g., CLSI). The provided research notes do not explicitly confirm this specific breakpoint for co-amoxiclav against Enterobacterales in CLSI/EUCAST.

The breakpoint aligns with CLSI standard. We have clarified and included this in the manuscript under Section 2.2 Microbiology as follows (lines 127-128):

Enterobacterales MICs for co-amoxiclav were determined by VITEK® 2 system, with MIC ≤ 8/4µg/mL considered as susceptible as per CLSI recommendation [17].”

- Significance of Non-Susceptible Isolates: The finding that 40.5% of Enterobacterales isolates were non-susceptible to amoxicillin (MIC > 8µg/mL) [lines 220-222] is a crucial piece of information. The discussion could further elaborate on how this high proportion of non-susceptible isolates might have influenced the observed clinical deterioration rates in both groups and reinforced the need for optimized dosing.

Thank you for pointing out this crucial piece of information. We have worded this into paragraph 3 of Section 4.1 Results interpretation and implications (lines 360-367).

“Notably, a significant 40.5% of Enterobacterales isolates in this study had amoxicillin MIC > 8µg/mL. Amongst patients with ≥1 Enterobacterale(s) isolated from their first intra-operative lower limb culture with amoxicillin MIC > 8µg/mL, there were fewer patients who experienced clinical deterioration in OG (6/23, 26.1%) compared to SG (20/27, 74.1%) (p<0.05). While use of IVCA at conventional dose of 1.2g Q8H for Enterobacterales with amoxicillin MIC > 8µg/mL has been reported to be predictive of therapeutic failure [14,16,30], our study suggests that empiric optimised IVCA dose of 1.2g Q6H reduces risk of clinical deterioration.”

Specific Comments

Introduction:

The phrasing regarding EUCAST and CLSI recommendations for the 1.2g every 6 hours dosing regimen (lines 69–71) should be revisited to ensure precision and alignment with the latest published guidelines.

Thank you for your comment on the need to revise the phrasing. We have rephrased the recommendation in the manuscript under the 5th paragraph of Section 1: Introduction (lines 66-83):

“Available pharmacokinetics/pharmacodynamics (PK/PD) data showed that IVCA 1.2g Q6H achieved a probability of target attainment of 90% (for PK/PD target of >40% time above MIC) for Enterobacterales with MIC of ≤ 4/2μg/mL [15-16]. Hence, based on the cur-rent susceptible breakpoint of ≤ 8/4µg/mL for co-amoxiclav towards Enterobacterales, Clinical and Laboratory Standards Institute (CLSI) recommends IVCA 1.2g Q6H [17]. In addition, EUCAST recommends IVCA 1.2g Q6-8H (standard dose) or IVCA 1.2g Q8H with IV amoxicillin 1g Q8H (high-dose), without specifying infection type [18]. Appropriate dosing of IVCA is crucial for DFI management, which has been associated with high rates of lower limb amputation (13.3 per 100 000 population in 2013) and 30-day mortality (11.1%) in Singapore [2]. Furthermore, patients with DFI tend to have compromised blood flow due to poor perfusion [2,6], which aggravates the concern of inadequate antibiotic penetration to infected lower limbs with conventional dose of IVCA 1.2g Q8H. At our institution, physicians are less likely to prescribe high-dose IVCA 1.2g Q8H with IV amoxicillin 1g Q8H upfront, due to lack of experience with such aggressive dosing regimen. Therefore, from August 2023, the antimicrobial stewardship unit (ASU) team revised the in-house antibiotic guidelines and recommended an optimised dose of IVCA at 1.2g Q6H for patients with actual body weight (ABW) > 50kg and creatinine clearance (CrCl) ≥ 50mL/min.”

Materials and Methods:

- Clarify the source/standard for the MIC breakpoint of ≤ 8/4µg/mL for Enterobacterales [lines 119-120].

The breakpoint aligns with CLSI standard. We have clarified and included this in the manuscript under Section 2.2 Microbiology as follows (lines 127-128):

Enterobacterales MICs for co-amoxiclav were determined by VITEK® 2 system, with MIC ≤ 8/4µg/mL considered as susceptible as per CLSI recommendation [17].”

Discussion:

- Refine the statement about "even higher dose" and EUCAST recommendations [lines 363-364] to ensure accuracy with published guidelines.

EUCAST defined high dose IVCA as 2g amoxicillin + 0.2 g clavulanic acid three times daily, equivalent to 6g of amoxicillin daily (Manuscript reference 18). We have worded into the 4th paragraph of Section 4.1 Results interpretation and implications, to improve clarity (lines 380-388).

“Amongst the 4 patients in OG who experienced clinical deterioration and had Enterobacterales with amoxicillin MIC 4-8µg/mL isolated from intra-operative lower limb culture, all had DFO, IWGDF/IDSA DFI grade 3 and CrCl >100mL/min. This suggests that the optimised dose of IVCA 1.2g Q6H might still be insufficient to achieve free co-amoxiclav levels above MIC for at least 40% of the dosing interval, an established PK/PD efficacy metric for β-lactams [16]. Perhaps, an even higher dose of amoxicillin (IVCA 1.2g Q8H with IV amoxicillin 1g Q8H), corresponding to a daily intake of 6g of amoxicillin, may be required as per high dose recommendation by EUCAST [18] in patients with deep -seated infections and good renal function.”

- While the safety data showed no significant difference, consider discussing if the study was powered to detect subtle increases in the incidence or severity of common adverse events like diarrhoea, given that previous literature suggests higher doses can increase severity.

Thank you for highlighting this. We acknowledge that our study was not powered to detect these subtle differences. As such, we have included the above points into our manuscript under the 5th paragraph of Section 4.1 Results Interpretation and Implications (lines 399-404).

“Incidence of severe adverse events due to IVCA is generally low, with hepatotoxicity occurring in 1 in every 2350 patients prescribed with co-amoxiclav [21]. Although our study was not powered to detect subtle differences in the incidence of adverse events, we could possibly conclude that use of IVCA 1.2g Q6H remains generally safe compared to IVCA 1.2g Q8H. Diarrhoea, which occurs in about 10% of patients in OG, was comparable to the incidence rate of other commonly prescribed antibiotics [34].”

Reviewer 2 Report

Comments and Suggestions for Authors

Overall, this is a well-designed and conducted single-center cohort study that evaluates an important clinical question regarding optimal dosing of intravenous co-amoxiclav for diabetic foot infections. I have some minor suggestion: 

Introduction: Provide a bit more background on the challenges and importance of appropriate antibiotic dosing for diabetic foot infections specifically.

Methods: Clarify the criteria used for diagnosing diabetic foot osteomyelitis.

Please, provide more details on how clinical deterioration was assessed (i.e. the factors considered beyond just worsening wound condition) and specify if any patients received concomitant antibiotics in addition to IV co-amoxiclav.

Results: it would be helpful to know the most commonly isolated pathogens.

Discussion: Expand the discussion on why higher doses may benefit even those with susceptible pathogens isolated.

It is possible the emergence of resistance with higher doses?

Author Response

Please see the attachment for PDF document of author's reply to the review report (Reviewer 2)

Introduction: Provide a bit more background on the challenges and importance of appropriate antibiotic dosing for diabetic foot infections specifically.

One of the key challenge for IVCA dosing was the resistance from primary physicians in the context of using a high dose IVCA 2.2g Q8H, due to unfamiliarity and concerns with side effects. We had to manage this resistance from the physicians by recommending an optimised dose of IVCA 1.2g Q6H instead (lines 73-83). Please refer to the 6th paragraph of Section 1: Introduction for details:

“Appropriate dosing of IVCA is crucial for DFI management, which has been associated with high rates of lower limb amputation (13.3 per 100 000 population in 2013) and 30-day mortality (11.1%) in Singapore [2]. Furthermore, patients with DFI tend to have compromised blood flow due to poor perfusion [2,6], which aggravates the concern of inadequate antibiotic penetration to infected lower limbs with conventional dose of IVCA 1.2g Q8H. At our institution, physicians are less likely to prescribe high-dose IVCA 1.2g Q8H with IV amoxicillin 1g Q8H upfront, due to lack of experience with such aggressive dosing regimen. Therefore, from August 2023, the antimicrobial stewardship unit (ASU) team revised the in-house antibiotic guidelines and recommended an optimised dose of IVCA at 1.2g Q6H for patients with actual body weight (ABW) > 50kg and creatinine clearance (CrCl) ≥ 50mL/min.”

Methods: Clarify the criteria used for diagnosing diabetic foot osteomyelitis.

Thank you for seeking clarity on the criteria for diagnosing DFO. In our institution, we use a combination of investigations as part of routine workup, including radiological tests of the affected foot (plain X-ray and/or MRI), intra-operative findings, probe-to-bone test and bone sample for cultures (Manuscript reference number 7, Page 5, Recommendations 7, 8 and 10). This clarification has been incorporated into the manuscript as follows (lines 112-117):

“Patients were deemed to have clinical diagnosis of diabetic foot osteomyelitis (DFO) if they fulfil ≥1 positive finding(s) from our institution’s routine workup for DFO, including: (i) radiological findings compatible with osteomyelitis (plain X-ray and/or magnetic resonance imaging of the foot); (ii) positive probe to bone test; (iii) intra-operative findings consistent with diagnosis of DFO; (iv) microbial growth in proximal bone chip cultures [7].”

Please, provide more details on how clinical deterioration was assessed (i.e. the factors considered beyond just worsening wound condition) and specify if any patients received concomitant antibiotics in addition to IV co-amoxiclav.

With regards to clinical deterioration, we can refer to Supplementary Table 2 (line 150) for a detailed explanation on how clinical deterioration was assessed. Aside from worsening wound condition, patients were deemed to have experienced clinical deterioration if they also fulfil ≥ 2 SIRS criteria, or if there were up-trending inflammatory markers (procalcitonin, CRP and white blood cell count).

With regards to empirical initiation of antibiotics for DFI treatment, none of our patients received concomitant antibiotics empirically in addition to IVCA according to hospital guidelines. We do not expect any of our patients to receive concomitant antibiotics as they do not have any nosocomial exposure (no hospitalisation or antibiotic exposure for past 3 months prior to admission for DFI). As such, they do not have risk factors for MRSA or Pseudomonas that will warrant the administration of concomitant IV vancomycin or IV ciprofloxacin/levofloxacin respectively. Patients who had been on antibiotic course in the last 3 months prior to the initiation of IVCA for treatment of DFI would have been excluded from the study (lines 96-97).

Some of our study patients may have concomitant antibiotics added to their regimen on availability of intra-operative lower limb culture results. These patients will have their clinical outcome (deterioration vs. improvement) assessed prior to the point of addition of concomitant antibiotic. This has been described under Section 2.4 Primary Efficacy Outcome (lines 142-145).

“For patients with concomitant antibiotics added during IVCA therapy (ie. IV vancomycin or PO clindamycin for MRSA isolated from their first intra-operative lower limb culture), the patient will be assessed for clinical deterioration prior to the addition of anti-MRSA agent.”

Results: It would be helpful to know the most commonly isolated pathogens.

The most commonly isolated pathogens were Staphylococcus and Streptococcus. We have included this statement, with the numerical breakdown and percentage of the pathogens, in the manuscript under Section 3.2 Microbiology (lines 231-236) as follows:

“A total of 156 out of 189 (75 in SG; 81 in OG) patients (82.5%) had organisms isolated in their first intra-operative lower limb culture result. The most commonly isolated pathogens were Staphylococcus (79 out of 323 isolates, 24.5%) and Streptococcus (57 out of 323 isolates, 17.6%). Of the 323 isolates, 145 (81 in SG; 64 in OG) were Gram-positive, 153 (86 in SG; 67 in OG) were Gram-negative, 20 (8 in SG; 12 in OG) were anaerobes and 5 (2 in SG; 3 in OG) were fungi”

Although Enterobacterales form the biggest share (126 out of 323 isolates, 39%), we did not list them as the most commonly isolated pathogens, since our Enterobacterales include pathogens from many different genera.

Discussion: Expand the discussion on why higher doses may benefit even those with susceptible pathogens isolated.

Thank you for the comment. We would like to highlight our discussion from lines 352-357, where higher doses may benefit those with susceptible Gram (+) pathogens isolated, and patients with susceptible Enterobacterales with very low amoxicillin MIC of ≤ 2µg/mL. This discussion was largely due to the lower observed rate of clinical deterioration in these patients in OG as compared to SG.

It is possible the emergence of resistance with higher doses?

We did not evaluate the possibility of resistance with optimised doses of IVCA. However, we think that optimised doses of IVCA will help to minimise the risk of antibiotic resistance instead. The discussion regarding resistance emergence has been incorporated into the manuscript under the 2nd paragraph of Section 4.2 Antimicrobial Stewardship (lines 435-439):

“Additionally, there is increasing evidence to suggest that optimised β-lactam dosing regimen maximises bacterial cell kill and minimises risk for β-lactam resistance due to subtherapeutic levels [36-38]. While antibiotic resistance was not evaluated in our study, we believe that optimising IVCA dosing is an antimicrobial stewardship strategy to retard emergence of resistance in DFI management.”

Reviewer 3 Report

Comments and Suggestions for Authors

This study comprehensively evaluated the treatment outcomes of 189 patients receiving high-dose IVCA. It was found that high-dose IVCA can effectively replace broad-spectrum antibiotics in clinical practice. The study also assessed the hepatotoxicity of high-dose IVCA in patients, which has reference significance for clinical application. However, there are still limitations to the use of high-dose IVCA. Therefore, the following comments are proposed.

  1. Regarding the abstract, it is too long. Please briefly describe the background and methods. Please emphasize the results section.
  2. Please note the format of the data units in the manuscript.
  3. Does the use of high doses of intravenous co-amoxiclav lead to the development of pathogen resistance? Recommendations for measures to address the possibility of drug resistance should be added to the manuscript.
  4. Does high-dose IVCA treatment cause toxicity in other organs? For example, renal toxicity. It is recommended to supplement the analysis of IVCA treatment toxicity to other human organs in order to comprehensively assess the side effects of high doses.
  5. The conclusion section should summaries the most important findings and add future prospects. Please revise the conclusion.
  6. I suggest the authors to check English for typos.
Comments on the Quality of English Language

I suggest the authors to check English for typos.

Author Response

Please see the attachment for PDF document of author's reply to the review report (Reviewer 3)

1. Regarding the abstract, it is too long. Please briefly describe the background and methods. Please emphasize the results section.

Thank you for pointing this out. We have adjusted the abstract as follows:

“Background: With increasing pharmacokinetic evidence suggesting inadequacy of conventional dose intravenous co-amoxiclav (IVCA) 1.2g Q8H in targeting Enterobacterales, our institution antibiotic guidelines optimised dosing recommendations for diabetic foot infection (DFI) management to 1.2g Q6H in August 2023. In this study, we aim to evaluate efficacy and safety of optimised dose IVCA in DFI treatment. Methods: In this single-centre cohort study, patients ≥21 years with DFI, creatinine clearance ≥50mL/min and weight >50kg, who were prescribed IVCA 1.2g Q8H (standard group (SG)) were compared with those prescribed IVCA 1.2g Q6H (optimised group (OG)). Patients who were pregnant, immunocompromised, had nosocomial exposure in last 3 months, or received <72 hours of IVCA were excluded. The primary efficacy outcome was clinical deterioration at end of IVCA monotherapy. The secondary efficacy outcomes include 30-day readmission and mortality, empiric escalation of antibiotics, lower limb amputation and length of hospitalisation. The safety outcomes include hepatotoxicity, renal toxicity and diarrhoea. Results: There were 189 patients (94 in SG; 95 in OG) included. Patients in SG (31.9%) were twice as likely to experience clinical deterioration compared to OG (16.8%) (odds ratio: 2.31, 95% confidence interval: 1.16-4.62, p<0.05). There were statistically more patients who had 30-day all-cause mortality in SG (5.3%) compared to OG (0%) (p<0.05). Further-more, 30-day readmission due to DFI in SG (26.6%) was higher compared to OG (11.6%) (p<0.05). Empiric escalation of IV antibiotics was required for 14.9% patients in SG and 6.3% patients in OG (p=0.06). There was no statistical difference for lower limb amputation (p=0.72), length of hospitalization (p=0.13) and occurrence of safety outcomes in both groups. Conclusion: This study suggests IVCA 1.2g Q6H is associated with decreased likelihood of clinical deterioration and is likely as safe as IVCA 1.2g Q8H. Optimised dose IVCA may help reduce use of broad-spectrum antibiotics due to clinical deterioration.”

2. Please note the format of the data units in the manuscript.

We note there has been a typo in the data unit for MIC of Enterobacterales with borderline amoxicillin susceptibility and have amended it from µg/L to µg/mL (line 350). Although the conclusion by Marti C et al. was “amoxicillin level of >40mg/L was predictive of occurrence of adverse events” (Manuscript reference 33, Page 5), we decided to change the data unit from 40mg/L to µg/mL, to ensure units are consistent in the context of antibiotic concentration level (line 393).

3. Does the use of high doses of intravenous co-amoxiclav lead to the development of pathogen resistance? Recommendations for measures to address the possibility of drug resistance should be added to the manuscript.

We did not evaluate the possibility of resistance with optimised doses of IVCA. However, we think that optimised doses of IVCA will help to minimise the risk of antibiotic resistance instead. The discussion regarding resistance emergence has been incorporated into the manuscript under the 2nd paragraph of Section 4.2 Antimicrobial Stewardship (lines 435-439):

“Additionally, there is increasing evidence to suggest that optimised β-lactam dosing regimen maximises bacterial cell kill and minimises risk for β-lactam resistance due to subtherapeutic levels [36-38]. While antibiotic resistance was not evaluated in our study, we believe that optimising IVCA dosing is an antimicrobial stewardship strategy to retard emergence of resistance in DFI management.”

4. Does high-dose IVCA treatment cause toxicity in other organs? For example, renal toxicity. It is recommended to supplement the analysis of IVCA treatment toxicity to other human organs in order to comprehensively assess the side effects of high doses.

In this study, we did not observe other side effects due to IVCA, including acute kidney injury (AKI as defined by KDIGO 2012) while receiving IVCA. The main focus of our study was on the efficacy of high-dose IVCA in treatment of antibiotic-naïve patients with DFI. The methodology for AKI has been included under the 3rd paragraph of section 2.6 Safety Outcomes (lines 187-192) and discussed under the 5th paragraph of section 4.1 Results interpretation and implications (lines 405-407).

“Renal toxicity, in the form of acute kidney injury, was defined based on the definition by the Kidney Disease: Improving Global Outcomes (KDIGO). Patients who developed either of the following were deemed to have acute kidney injury within 30 days of IVCA initiation: (a) rise in serum creatinine by at least 50% from baseline; (b) absolute increase in serum creatinine by ≥ 0.3mg/dL; (c) urine output < 0.5mL/kg/h for ≥ 6 consecutive hours [28].”

“None of the patients in both groups developed other types of toxicity (including acute kidney injury) attributed to IVCA.”

5. The conclusion section should summaries the most important findings and add future prospects. Please revise the conclusion.

We have revised the conclusion to summarise the most important findings as follows:

  • High-dose IVCA 1.2g Q6H associated with reduced likelihood of clinical deterioration
  • High-dose IVCA 1.2g Q6H is likely as safe as IVCA 1.2g Q8H
  • High-dose IVCA 1.2g Q6H may reduce escalation to broader-spectrum antibiotics beyond IVCA when there is clinical deterioration

We have also included possible future studies on the use of higher doses of IVCA in our conclusion. You may refer to Section 5: Conclusion, lines 475-482.

“Optimised dose IVCA 1.2g Q6H is associated with reduced likelihood of clinical deterioration and is likely as safe as conventional dose IVCA 1.2g Q8H, in the treatment of DFI. Optimised dose IVCA 1.2g Q6H may help reduce use of even broader spectrum antibiotics beyond co-amoxiclav, via antibiotic escalation due to clinical deterioration. Further studies on higher doses of amoxicillin (IVCA 1.2g Q8H with IV amoxicillin 1g Q8H) for patients who have deep-seated DFI and augmented renal clearance; or IVCA 1.2g Q6H for patients with deep-seated DFI and suboptimal renal function of CrCl < 50mL/min) are required to determine if they are safe and effective for this patient population.”

6. I suggest the authors to check English for typos.

We have proofread the entire manuscript and made changes for typos or grammatical errors. We have also changed some of the words in the manuscript to reflect the British English spelling as per MPDI requirements.

Round 2

Reviewer 3 Report

Comments and Suggestions for Authors

The revised version of the manuscript may be considered for publication.